# Controlling light emission from semiconductor nanoplatelets using surface chemistry

Michael W. Swift [1] ✉, Alexander L. Efros [1] ✉ & Steven C. Erwin [1] ✉

Semiconductor nanoplatelets are atomically flat nanocrystals which emit light with high spectral purity at wavelengths controlled by their thickness. Despite their technological potential, efforts to further sharpen the emission lines of nanoplatelets have generally failed for unknown reasons. Here, we demonstrate theoretically that the linewidth is controlled by surface chemistry—specifically, inhomogeneities in the ligand layer on the nanoplatelet surface lead to a spatially fluctuating potential that localizes excitons. This localization leads to increased scattering and optical broadening. Importantly, localization also reduces the rate of radiative emission. Our model explains the observed linewidth and predicts that a more uniform ligand layer will sharpen the lines and increase the emission rates. These findings demonstrate that light emission from nanoplatelets can be controlled by optimizing their surface chemistry, an important advantage for their eventual use in optical technologies.

Semiconductor nanoplatelets are extremely thin nanocrystals consisting of just a few atomic layers. This thickness can be controlled at the atomic level, enabling the precise manipulation of photon emission and absorption over a wide spectral range of more than one eV[1–14]. Moreover, the Coulomb interaction between electrons and holes, which is strongly enhanced by their two-dimensional confinement as well as by the small dielectric constant of the surrounding medium, leads to exciton binding energies that can reach hundreds of meV. Consequently, the photoluminescence from nanoplatelets is generally stable even at room temperature. In addition to this tunability and stability, nanoplatelets exhibit extremely fast radiative recombination —with lifetimes as short as a few picoseconds—and hence high brightness and efficiency[2,15–18]. For all these reasons, nanoplatelets have the potential to become the most effective colloidal luminophores among current nanomaterials.

Notwithstanding these advantages, a problem remains: the optical linewidths of the emission spectra in nanoplatelets are unexpectedly broad—for CdSe nanoplatelets, in the range 35 to 55 meV at room temperature[1,2,7,13,19,20] and 10–20 meV at cryogenic temperature[7,18,21]. Other semiconductor nanoplatelets, such as mercury chalcogenides, show similar behavior[22]. While these are fairly narrow linewidths compared to ensembles of three-dimensional nanocrystals, applications such as LEDs and lasers would benefit from even sharper emission[23,24]. Broadening due to phonon coupling is likely unavoidable at room temperature. However, the substantial linewidths at cryogenic temperatures indicate that there are also temperature-independent broadening mechanisms at work, which may be possible to mitigate. Shape and size dispersion lead to temperature-independent broadening in ensembles of three-dimensional nanocrystals. This mechanism cannot explain the broadening in nanoplatelets: first, because nanoplatelet ensembles are routinely grown with nearly zero dispersion in their thickness; second, because the room-temperature linewidths from individual nanoplatelets are very similar to those of ensembles[25–28]. In light of these facts it is clear that the broadening mechanism must be a property of individual nanoplatelets[18]. Because the origin of broadening in nanoplatelet emission is not currently known, experimental efforts to further sharpen the lines have been unsuccessful.

The same mechanism that broadens emission lines is likely also responsible for the fact that recombination rates in nanoplatelets, although fast, are slower than estimated theoretically[2]. This is generally true because emission linewidths for weakly confined excitons are

[1]Center for Computational Materials Science, Naval Research Laboratory, Washington, DC, USA. ✉e-mail: michael.w.swift5.civ@us.navy.mil; alex.l.efros.civ@us.navy.mil; steven.c.erwin.civ@us.navy.mil

proportional to lifetimes[29]. In the case of nanoplatelets, the observed linewidths are much larger than linewidths expected from intrinsic lifetime broadening or typical fluctuations in lateral size (Fig. 1c), suggesting that the excitons are localized in small regions by some kind of disorder[18,30]. This also reduces the rate of radiative recombination, much like in the case of ideal quantum wells: reduction of the area of coherent motion slows recombination[31] (Fig. 1d). If the nature of this disorder can be identified and eliminated, the recombination rate will thus be increased.

In this work, we propose that nanoplatelet surface chemistry—specifically, spatial inhomogeneities in the ligand layer used to passivate the nanoplatelet surface—is responsible for broadening emission and slowing recombination. Fluctuations in the ligand layer yield a spatially fluctuating potential that localizes the exciton. This broadens the photoluminescence and reduces the recombination rate of the bright state. We develop here a microscopic model (implemented in Python; see Suppl. Data 1) that describes these effects quantitatively. The model only depends on a single parameter, $\alpha$, that captures the shift in local band gap because of the fluctuation. It can, therefore, be applied to any fluctuation mechanism, provided that an appropriate $\alpha$ can be determined.

## Results

### Light emission from ideal nanoplatelets

The electron and hole wavefunctions $\chi_e$ and $\chi_h$ for a thin nanoplatelet, in the direction perpendicular to the surface, may be approximated by the lowest-lying bound states of an infinite square well with width equal to the nanoplatelet thickness $d$. The corresponding exciton is strongly confined in the direction perpendicular to the surface while its in-plane motion is nearly free. In this weakly confined regime the exciton radius is much smaller than the lateral size of the nanoplatelet, allowing for a separation of variables into the relative coordinate $\boldsymbol{\rho}$ and center-of-mass coordinate $\mathbf{R}$. The exciton wavefunction then has the form

$$\Psi_{ex}(\mathbf{r}_e,\mathbf{r}_h) = \chi_e(z_e)\chi_h(z_h)\phi_{2D}^d(\boldsymbol{\rho})\psi(\mathbf{R}). \quad (1)$$

The relative wavefunction $\phi_{2D}^d$ is found variationally (section "Wavefunction of relative motion")[32]. In an ideal nanoplatelet—one with a perfectly flat and uniform surface—the center-of-mass wavefunction $\psi$ is the lowest-lying bound state of an infinite potential well in the shape of the nanoplatelet (Fig. 1a).

From the wavefunction of the exciton we can calculate its lifetime $\tau_{ex} = \tau_0/\mathcal{D}^2 K$, where $\tau_0$ is a characteristic lifetime, $\mathcal{D}$ is the depolarization factor, and $K$ is the square of the overlap integral[33]. The characteristic lifetime of a nanoplatelet depends on its thickness because it varies with the frequency of the emitted light. In CdSe nanoplatelets, $\tau_0$ varies with thickness from 0.5 to 0.8 ns (Suppl. Fig. 5). The depolarization factor describes the reduction of the electric field inside the nanostructure due to the dielectric discontinuity at the surface. In a nanoplatelet, emission is typically polarized in the plane. Since the dielectric discontinuity only affects the normal component of the field, $\mathcal{D}^2 = 1$. For comparison, $\mathcal{D}^2 \approx 0.4$ in a quasi-spherical CdSe nanoparticle[33]. The square of the overlap integral is:

$$K = \left| \int d^3\mathbf{r}_e d^3\mathbf{r}_h \Psi_{ex}(\mathbf{r}_e,\mathbf{r}_h)\delta(\mathbf{r}_e - \mathbf{r}_h) \right|^2. \quad (2)$$

For ideal rectangular nanoplatelets with edge lengths $L_x$ and $L_y$, we evaluate $K$ to find,

$$\frac{1}{\tau_{ex}} = \frac{1}{\tau_0}\frac{128}{\pi^4}\frac{L_x L_y}{\pi a_{2D}^2(d)} \approx \frac{1.31}{\tau_0}\frac{S}{\pi a_{2D}^2(d)}, \quad (3)$$

where $S = L_x L_y$ is the nanoplatelet surface area and $a_{2D}$ is the exciton radius (found in the section "Wavefunction of relative motion"). We

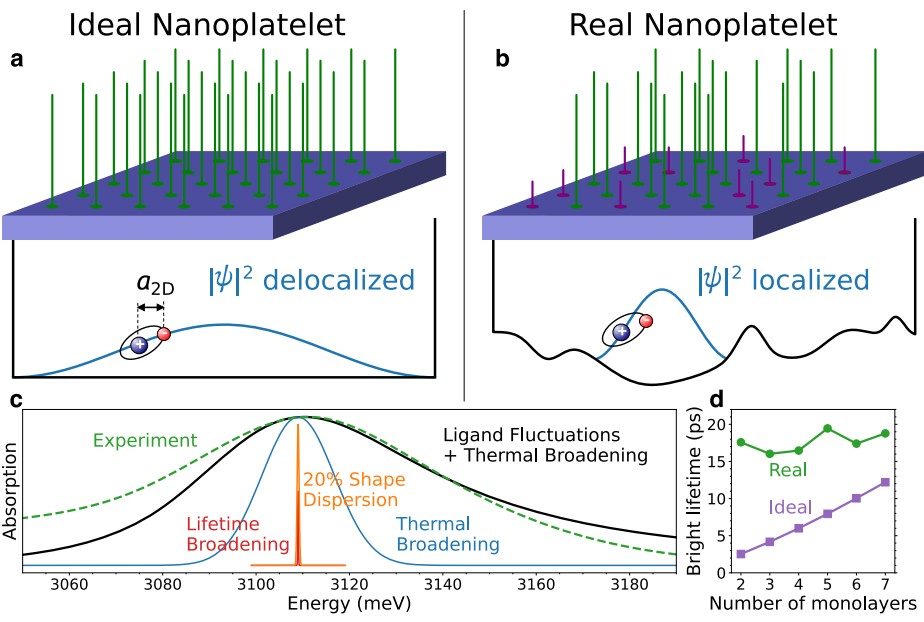

**Fig. 1 | Exciton behavior in ideal and real nanoplatelets.** Comparison of exciton behavior in ideal nanoplatelets, for which every surface site is passivated by the same ligand, to real nanoplatelets, for which two different ligands randomly occupy the surface binding sites. **a** In ideal nanoplatelets the exciton center of mass moves coherently over the whole nanoplatelet. This leads to extremely narrow linewidths (of order 1 meV) and very short radiative lifetimes (<10 ps) for the bright states, in strong disagreement with experiments. **b** In real nanoplatelets, spatial fluctuations in the ligand layer lead to a rougher energy landscape. This effect localizes the

excitons, leading to much larger linewidths of 30–60 meV (**c**) and longer bright-state lifetimes of 15–20 ps (**d**), in much better agreement with experimental data. The theoretical results shown here are for square 30-nm CdSe nanoplatelets. In panel **c** the predicted lineshape (black curve) is for a 2-monolayer platelet, including both ligand fluctuations and thermal broadening, while the experimental lineshape is for 2-monolayer platelets at room temperature from ref. 19. Source data are provided as a Source Data file.

used this expression to calculate the bright state lifetimes in Fig. 1d. Note that the radiative rate is faster than the characteristic rate $1/\tau_0$ by a geometric factor equal to the ratio of the nanoplatelet area to the exciton area, a phenomenon known as "giant oscillator strength". The exciton radius in a nanoplatelet is comparable to its thickness (Suppl. Fig. 1). That means the ratio $S/\pi a_{2D}^2(d)$ is very large, typically of order $10^2$ or more. Hence, the lifetime of the optically active state in ideal nanoplatelets can be extremely short, of order 5–10 ps (Fig. 1d).

### Broadening of light emission from real nanoplatelets

Exciton recombination rates measured experimentally are indeed fast, as expected from their giant oscillator strength. However, the room-temperature emission has an unexpectedly large linewidth (in the range 35–55 meV) that remains unexplained. As mentioned above, even single nanoplatelets show these linewidths, and thus we can rule out variations in their lateral size as a broadening mechanism.

We note that emission is broader at room temperature than at cryogenic temperature, indicating that phonon coupling is an additional source of broadening. Furthermore, single nanoplatelets at cryogenic temperatures have achieved nearly indistinguishable single-photon emission, with linewidths of approximately 1 meV limited by the lifetime[26–28]. This occurs when the exciton population is funneled into the single lowest exciton state in a nanoplatelet, rather than thermalized over a distribution of states. Our model focuses on absorption lineshapes, which are much simpler because they are independent of temperature and do not have population funneling effects. Emission and absorption linewidths should be similar at room temperature, and certainly whatever broadens emission will broaden absorption.

Our proposed resolution to the linewidth puzzle is motivated by semiconductor alloys, where spatial compositional fluctuations lead to broadening of exciton lines. In some cases the exciton extends over many lattice sites and experiences an effective band gap given by the average alloy content. But if the fluctuations are sufficiently strong they can lead to distinct regions with larger or smaller average band gap. Excitons can be trapped in small-gap regions or scatter off large-gap regions. Both effects broaden the emission lines[34,35].

We propose that a similar mechanism occurs in semiconductor nanoplatelets: spatial fluctuations in the type of surface ligands can mimic alloy compositional fluctuations by modifying the effective band gap on the scale of individual atomic sites. The fluctuations in the local average energy gap, in turn, localize or scatter the excitons and—most significantly—substantially broaden the optical emission lines and reduce the emission rate. We present our model in detail below and explore its consequences for the light emitted by exciton recombination in nanoplatelets.

We describe the effect of ligand fluctuations as a random potential $V(\mathbf{R})$ that acts on the exciton center of mass. We consider the resulting exciton states, indexed by $i$. The exciton wavefunctions still have the form given in Eq. (1), but the center-of-mass wavefunctions $\psi_i(\mathbf{R})$ are modified by the random potential. For example, the exciton may be localized in a region where fluctuations in the ligand layer lead to a slightly reduced average band gap, as sketched in Fig. 1b. The radiative recombination rate of state $i$ is $1/\tau_i = K_i/\tau_0$, where $K_i$ is the overlap integral squared. Evaluating the overlap and then averaging over the states we obtain the average recombination rate at energy $\varepsilon$,

$$\frac{1}{\tau(\varepsilon)} = \frac{|\phi_{2D}^d(0)|^2}{\tau_0}\frac{A(\varepsilon)}{n(\varepsilon)}, \tag{4}$$

where $n(\varepsilon) = \sum_i \delta(\varepsilon - \varepsilon_i)$ is the density of states and $A(\varepsilon) = \sum_i |\int d^2\mathbf{R}\,\psi_i(\mathbf{R})|^2 \delta(\varepsilon - \varepsilon_i)$ is proportional to the probability of absorbing a photon with energy $\varepsilon$[34,35]. Hence, $A(\varepsilon)$ may be thought of as describing the absorption lineshape, so its full-width at half-maximum corresponds to the absorption linewidth.

The assumed form of the random potential and the resulting functions $A(\varepsilon)$ and $n(\varepsilon)$ are in the section "Ligand fluctuation model". They depend on three parameters: the fraction $x$ of surface sites passivated by the first type of ligand (the second type passivates the remaining fraction $1 - x$); the areal density of ligand sites $N$; and the rate of change of the band gap with ligand passivation, $\alpha = dE_g/dx$. These parameters can be combined to define a natural energy scale $W = x(1 - x)\alpha^2 M/2\pi\hbar^2 N$ that characterizes the typical potential fluctuations, where $M = m_e + m_h$ is the effective mass of the exciton center of mass motion. The effects of the random potential are proportional to this energy scale, so we can use it to define dimensionless quantities and examine the universal behavior abstracted from changes in physical parameters. In particular, we define the dimensionless energy $\varepsilon^* = \varepsilon/W$ and the dimensionless absorption lineshape $A^*(\varepsilon^*) = A(\varepsilon)W/S$. Additionally, since $A(\varepsilon)/n(\varepsilon)$ is proportional to the radiative lifetime of the exciton state with energy $\varepsilon$[34,35], we define the dimensionless radiative lifetime $\tau^*(\varepsilon^*) = A^*(\varepsilon^*)n_0/n(\varepsilon^*)$, where $n_0 = SM/2\pi\hbar^2$ is the free-particle density of states.

The prediction of our model for the absorption lineshape $A^*(\varepsilon^*)$ and radiative lifetime $1/\tau^*(\varepsilon^*)$ reveal the universal effect of ligand fluctuations for infinite semiconductor nanoplatelets; see Fig. 2a. This result has three important implications. First, the full width at half maximum of $A^*(\varepsilon^*)$ is 4.04. This implies the linewidth due to

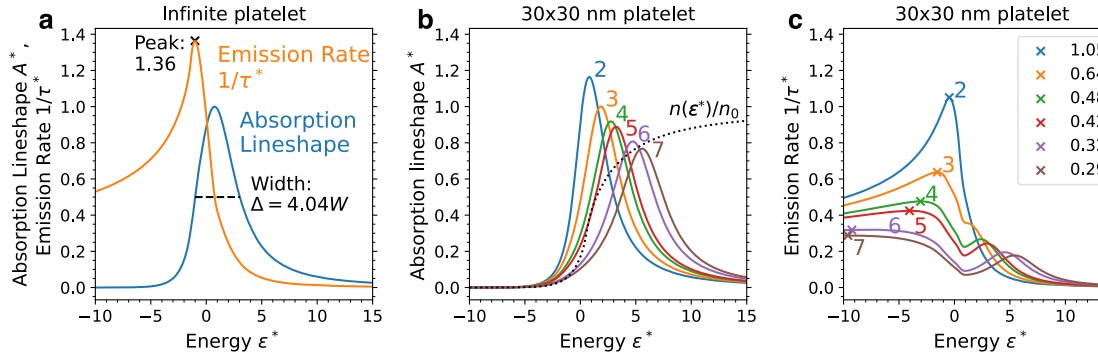

**Fig. 2 | Dimensionless excitonic lineshapes and rates. a** Dimensionless absorption lineshape $A^*$ (blue, here with peak height normalized to 1) and dimensionless radiative recombination rate $1/\tau^*$ (orange) obtained from optimal fluctuation theory for an infinite platelet. These quantities are plotted as a function of dimensionless energy $\varepsilon^*$. **b** $A^*$ and **c** $1/\tau^*$ for 30 nm by 30 nm CdSe platelets. The lines are color coded by thickness in monolayers (2 through 7) as labeled on the plot. The peak value of $1/\tau^*$ is shown in panel **c** by an "x", with the peak value labeled in the legend. Source data are provided as a Source Data file.

broadening by the ligands is given by

$$\Delta \approx 4.04 \frac{x(1-x)\alpha^2}{N} \frac{M}{2\pi\hbar^2}. \quad (5)$$

This expression reveals that the broadening is largest when the ligand mixing $x$ is 50%. In the following section we evaluate the maximum broadening for CdSe nanoplatelets and show the range of typical values is several tens of meV, which is consistent with numerous experimental measurements.

Second, the dimensionless recombination rate peaks at a value of 1.36. The maximum rate (minimum lifetime) is therefore given by

$$\frac{1}{\tau_{min}} \approx 34.5 \frac{\hbar^2 |\phi_{2D}^d(0)|^2}{M\tau_0} \frac{1}{\Delta}. \quad (6)$$

Though it may seem counterintuitive that recombination time is proportional to the linewidth, this is a general characteristic of weakly confined excitons[29].

Third, the recombination rate drops rapidly at higher energies. Above a certain temperature, the average exciton lifetime increases with increasing temperature as more of the slow states in the high-energy tail become populated. High-energy states become dimmer as they become similar to plane-wave states, which have absorption coefficients inversely proportional to the square of the energy[35]. As a result, the average lifetime will increase as temperature increases and higher-energy slower states become populated. Furthermore, the radiative lifetime is always longer in the presence of fluctuations (Suppl. Note 5 and Suppl. Fig. 13). Our model thus explains both the broadening and the reduced rate of emission.

So far, we have assumed a nanoplatelet that is infinite in lateral size. Once the effects of lateral confinement are taken into account (section "Confined fluctuations"), shifts from the universal behavior emerge when the size of the optimal fluctuations becomes comparable to the size of the nanoplatelet. The qualitative trends from the infinite case still provide useful intuition, but the effects of lateral confinement must be considered for quantitative accuracy in platelets that are 3 monolayers and thicker (Fig. 2b, c).

### Ligand fluctuations in cadmium selenide nanoplatelets

In CdSe nanoplatelets, several mechanisms can modify the effective gap. These mechanisms include the mixing of different types of ligands[36–38], such as short-chain and long-chain ligands[39], as well as a combination of organic and inorganic ligands[40]. The random arrangement of these ligands can lead to spatial fluctuations in local strain across the nanoplatelet. Mechanical deformation, curling, and differential strain between the edges and centers of the nanoplatelets are other factors that may cause similar fluctuations[41–45].

Modeling these strain effects comprehensively is challenging. The complexities involved in simulating nanoplatelets with various passivating ligands require large supercells to accommodate the ligand layer's preferred conformation. Additionally, curvature and edge effects demand unfeasibly large simulation cells. Furthermore, capturing the vibrations and rotations of ligand chains at finite temperatures would require molecular dynamics simulations. Given these complexities, a detailed investigation of these effects is beyond the scope of this work.

For our purposes, we focus on obtaining a reasonable estimate based on variations in the ligand shell by simulating a variety of small passivating ligands, such as acetate, chlorine, iodine, and sulfur, and using a mixture of these ligands as a proxy for ligand disorder. A rough estimate of the strain effects from short- and long-chain ligand disorder (see below) shows that they are comparable in size, so this proxy is reasonable.

We now apply the ligand fluctuation model to CdSe nanoplatelets. We start by using density-functional theory (DFT) to calculate the band gap, $E_g^\ell(d)$, for a nanoplatelet passivated by ligand $\ell$. Four ligands are considered: acetate (a short-chain carboxylate organic ligand) and three inorganic ligands: chlorine, iodine, and sulfur. The results are shown in Fig. 3a and additional details are in the section "Density-functional theory". The calculated gaps for $d = 4$ monolayers are in very good agreement with the experimental absorption onset energies; see Fig. 3b.

We define the optical shift for a nanoplatelet passivated with ligands $\ell$ and $\ell'$ to be the difference $\Delta E_g^{\ell,\ell'}(d) = E_g^\ell(d) - E_g^{\ell'}(d)$ between the DFT gaps of the passivated nanoplatelets; both energies depend on the thickness $d$ of the nanoplatelet[8]. The calculated shifts are shown in Fig. 3c as a function of the thickness $d$, for $\ell$ = acetate and $\ell'$ an inorganic ligand. When calculating lifetimes we will use experimental values for the photon energy (see Suppl. Fig. 5), since the DFT gaps do not include excitonic effects and may not fully capture confinement effects. However, these discrepancies should cancel when comparing DFT gaps for two different platelets with the same thickness but different ligands, so $\Delta E_g$ values should correspond closely to the true shifts in the photon energies.

We will focus on the comparison between passivation by acetate and sulfur. The 2D density of ligand sites is $N = 2/a_c^2$, where $a_c = 0.608$

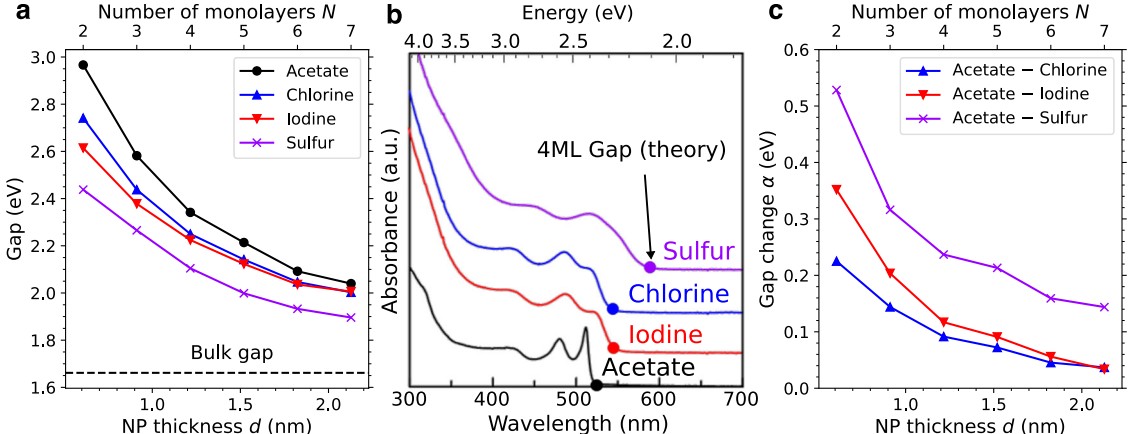

**Fig. 3 | Effect of ligand passivation on nanoplatelet band gap.** Optical transition energies of CdSe nanoplatelets passivated by different ligands: **a** DFT gaps between the valence and conduction states in CdSe nanoplatelets, comparing nanoplatelets that are passivated with different ligands ($E_g^\ell(d)$). **b** Comparison of theory with experiment. Theoretical gaps for the passivated nanoplatelets are shown as dots superimposed on the experimental absorbance data[8]. We observe excellent agreement between the calculated gaps and the experimental absorption onset. **c** Band gap shift parameter $\alpha = \Delta E_g^\ell(d)$ for the different ligands as a function of nanoplatelet thickness. Source data are provided as a Source Data file.

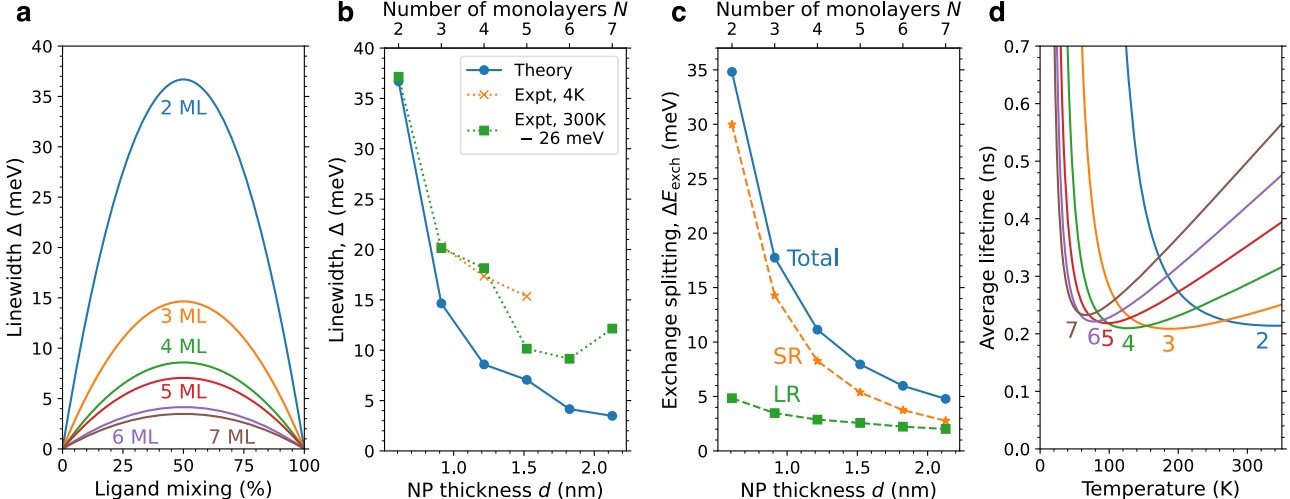

**Fig. 4 | Calculated excitonic properties of CdSe nanoplatelets. a** Predicted absorption linewidth of CdSe nanoplatelets passivated with a mix of acetate and sulfur as a function of ligand mixing for a range of nanoplatelet thicknesses (shown in monolayers, ML). **b** Maximum absorption linewidth (corresponding to 50 percent acetate and 50 percent sulfur) is shown in blue as a function of nanoplatelet (NP) thickness. Experimental linewidths are plotted for comparison. Linewidths at cryogenic temperatures from ref. 7 are shown in orange. Room-temperature experiments show thermal broadening, which is not included in our calculations. Therefore, for the sake of comparison, thermal broadening has been removed from

the room-temperature experimental data by subtracting $k_B T \approx 26$ meV from the linewidths (data from refs. 19,20, shown in green). **c** Exciton dark-bright exchange splitting in acetate-passivated CdSe nanoplatelets as a function of nanoplatelet thickness $d$. The short-range contribution is shown in orange, the long-range in green, and the total in blue. **d** Temperature-dependent radiative lifetime, $\tau_{avg}(T)$, of CdSe nanoplatelets. Each curve represents a different nanoplatelet thickness, labeled by the number of monolayers. Source data are provided as a Source Data file.

nm is the lattice constant of cubic CdSe[2]. Assuming every site is passivated by either acetate or sulfur, the fraction of acetate ligands is $x$ and the fraction of sulfur is $(1 − x)$. Assuming the effective band gap is linear with $x$, the rate of change of the band gap with ligand passivation is $\alpha = dE_g/dx = \Delta E_g^{\ell,\ell'}(d)$.

For the remainder of this work, we will use acetate-sulfur disorder as a proxy for any type of disorder in the ligand layer. For nanoplatelets passivated with only carboxylate ligands, the disorder comes in the form of variation in the ligand chain length—for example between acetate (a short-chain ligand) and oleate (a long-chain ligand). To justify the claim that ligand fluctuations can explain the linewidth in purely carboxylate-passivated nanoplatelets, we make a rough estimate of the effect of chain-length disorder on the band gap. Experimentally, oleate passivation for 4-monolayer platelets has been shown to lead to an average in-plane lattice constant of 0.619 nm[38]: 1.8% tensile strain compared to the bulk lattice constant of 0.608 nm. We performed a DFT calculation of 4-monolayer acetate-passivated nanoplatelets with full relaxation of all lattice parameters, and found acetate passivation leads to very small compressive biaxial strain: <0.2%. The variation in the band gap between acetate- and oleate-passivated platelets can, therefore, be estimated by comparing the CdSe band gaps under no strain and under 1.8% biaxial tensile strain.

We make this comparison using hybrid DFT, as shown in Suppl. Fig. 11. With the in-plane lattice parameter fixed to 0.619 nm, the relaxed out-of-plane lattice parameter is 0.588 nm, in good agreement with 0.594 nm measured for 4-monolayer platelets in ref. 38. The band gap of CdSe is particularly sensitive to biaxial tensile strain, since it breaks the threefold degeneracy of the valence band: one band moves up in energy while the other two move down. The band gap corresponding to an in-plane lattice parameter of 0.619 nm is 1.52 eV, compared to the bulk gap of 1.66 eV, for a gap shift of $\alpha = 0.14$ eV. This is comparable to, though somewhat smaller than, the value of $\alpha = 0.237$ eV associated with acetate-sulfur ligand disorder (Fig. 3c). However, the nominally oleate-passivated platelets in ref. 38 likely have some remaining acetate ligands, so the experimentally derived value of 0.619 nm is likely an underestimate of the average lattice

parameter in a purely oleate-passivated platelet. Furthermore, this estimate neglects the fact that the strain induced by long-chain ligands is anisotropic. Due to the zincblende structure, the top and bottom surfaces are rotated by 90° with respect to one another, so the directions of strongest tensile strain are orthogonal between the top and bottom surfaces[44]. The experimental diffraction measurements average out this anisotropy and therefore underestimate the degree of distortion due to the ligands. These considerations suggest that $\alpha = 0.14$ eV is an underestimate of the true broadening due to chain-length disorder: chain-length and ligand-type disorder will likely have a similar net effect on the linewidth. Since a detailed treatment of chain-length disorder is beyond the scope of this work, we focus on ligand-type disorder as a proxy for all ligand disorder.

Using our model, we calculated the exciton linewidths and lifetimes in CdSe nanoplatelets taking ligand fluctuations into account. Figure 4a shows how the linewidth depends on the acetate fraction $x$. Typical potential fluctuations that lead to this broadening have varying size and strength depending on the nanoplatelet thickness (Suppl. Fig. 6). One can see that the linewidth peaks for an even mix in ligand types, but it is substantial even at fairly large $x$, and noticeable narrowing should be possible even from modest improvements in ligand uniformity. The linewidth also depends on the nanoplatelet thickness because the exciton effective mass $M$ increases with thickness (Suppl. Note 4) and $\alpha$ decreases with thickness. These and many other calculated thickness-dependent quantities for CdSe nanoplatelets are in Suppl. Table 2.

Figure 4b compares the calculated maximum linewidth to experimental linewidths from refs. 7,19,20. Since our model does not include the effects of phonon coupling, thermal broadening must be taken into account when comparing with room-temperature linewidths, while the linewidths measured at cryogenic temperatures may be compared directly. We apply a simple heuristic to remove the effect of thermal broadening from the room-temperature experimental linewidths: subtract $k_B T$ (approximately 26 meV). This simple approach is validated by the good match between the experimental cryogenic linewidths and the adjusted room-temperature linewidths.

Many experiments have reported a strong dependence of linewidths on growth and processing conditions[46–48]. In light of our model, this dependence can now be rationalized as arising from differences in the ligand layers.

The optimal fluctuation model predicts the lifetime of the "bright" exciton, i.e., the state whose radiative recombination is dipole allowed. To compare with experimentally measured average lifetimes we must also consider the dipole-forbidden "dark" state, which is lower in energy (typically by 10–20 meV) in CdSe nanocrystals[49]. The populations of the bright and dark states are given by a Boltzmann distribution. Consequently, the temperature-dependent average lifetime depends sensitively on the exchange splitting between bright and dark levels, $\Delta E_{\text{exch}}$, which can be separated into short- and long-range components[14,50–52] as calculated in Suppl. Note 2 and shown in Fig. 4c.

We note that our calculated exchange splitting is significantly larger than earlier theoretical results[7]; see Suppl. Note 2D for further discussion. We further note that our calculated dark-state energy is in remarkable agreement with the energy of the low-lying peak observed in photoluminescence spectra of CdSe nanoplatelets below 160 K[7,26,39,53–56], raising the intriguing possibility that this controversial lower peak may be due to longitudinal-acoustic-phonon-assisted recombination[57,58] of the dark exciton (Suppl. Note 2D and Suppl. Fig. 8). While the evidence is far from conclusive, we believe this question merits further investigation.

With $\Delta E_{\text{exch}}$ in hand, we can now describe how the exciton population is spread across both the bright states [with density $n(\varepsilon)$] and dark states [with density $n(\varepsilon + \Delta E_{\text{exch}})$] according to a Boltzmann distribution. As discussed in the section "Radiative decay time", Eq. (4) allows us to calculate the average decay rate of the exciton at temperature $T$:

$$\frac{1}{\tau_{\text{avg}}(T)} = \frac{|\phi_{\text{2D}}^d(0)|^2}{\tau_0} \frac{\int A(\varepsilon) \exp(-\varepsilon/k_{\text{B}}T)d\varepsilon}{\int [n(\varepsilon) + n(\varepsilon + \Delta E_{\text{exch}})] \exp(-\varepsilon/k_{\text{B}}T)d\varepsilon}. \quad (7)$$

The resulting average exciton lifetime is shown as a function of temperature and nanoplatelet thickness in Fig. 4d. The lifetimes are in the sub-nanosecond regime and have a non-monotonic dependence on temperature. At low temperatures, the exciton population is concentrated in the dark state so recombination is very slow, but it speeds up with increasing temperature as the population of the bright exciton increases. As temperature increases further, emission again slows due to population of the higher-energy exciton states, which have rapidly decreasing emission rates (Fig. 2). The minimum lifetime for each thickness and the temperature $T_{\tau\,\text{min}}$ at which that lifetime is achieved are in Suppl. Table 2.

Experimental average lifetimes vary from 2.5 ns to 11 ns at room temperature[19,20], and are 10s to 100s of picoseconds at cryogenic temperatures[2,7,26,59]. Room-temperature lifetimes are underestimated by our model. This is due in part to our use of hard-wall boundary conditions when calculating the 2D exciton radius, which leads to an under-predicted exciton radius and hence an under-predicted lifetime. Additionally, we have neglected phonon-induced decoherence of the localized exciton states, which will increase the radiative decay time. Though careful treatment of this effect is beyond the scope of this work, a rough phenomenological model (Suppl. Note 6) shows that this mechanism can explain the discrepancy with experimental lifetimes.

## Discussion

Semiconductor nanoplatelets offer great promise as optical materials. They are optically bright, due to short radiative lifetimes and high quantum yields, and they can be readily grown monodisperse in their thickness, resulting in well-separated emission bands. One vexing issue has long remained unresolved: their emission linewidths are much larger than expected from known sources of broadening.

In this work, we proposed and demonstrated theoretically that a previously unsuspected source of broadening—spatial fluctuations in the ligands on the nanoplatelet surface—resolves this issue: the broadening arises from ligand-induced spatial fluctuations in the effective band gap felt by the excitons. We developed a physically transparent, quantitative model (implemented in Python; see Suppl. Data 1) that elucidates this effect. By applying our model to CdSe nanoplatelets we showed that physically plausible fluctuations in the ligand layers quantitatively account for the linewidths observed experimentally. This broadening occurs at the level of individual nanoplatelets and so even ensembles of identical nanoplatelets would show the same broad emission. Therefore, efforts to further reduce size and shape dispersion in the ensembles are unlikely to lead to sharper emission lines. Improved passivation of the surface is a more promising route toward improvement of nanoplatelets' optical properties. If passivation of surface sites by a uniform ligand shell could be achieved, a substantial improvement in linewidth might be obtained.

More broadly, our work suggests that optical emission in other nanocrystal quantum dot materials—not just nanoplatelets—is also likely to be strongly influenced by ligand fluctuations. More work is needed to directly investigate the influence of surface chemistry on optical properties and then to harness it. Our model highlights the importance of considering the interplay between physics and chemistry in the interdisciplinary field of nanotechnology. Careful characterization and control of the ligand chemistry may someday unlock the full potential of light emission from colloidal nanocrystals.

## Methods

### Wavefunction of relative motion

To determine the wavefunction of relative motion, we follow refs. 52 and 14 and find the exciton radius variationally using the Hanamura potential[32] to describe dielectric confinement, with the thickness-dependent exciton reduced mass $\mu(d) = m_e(d)m_h/(m_e(d) + m_h)$. We use the ansatz

$$\phi_{\text{2D}}^d(\rho) = \frac{\sqrt{2}}{a_{\text{2D}}(d)\sqrt{\pi}} \exp\left(-\frac{\rho}{a_{\text{2D}}(d)}\right). \quad (8)$$

The 2D exciton radius $a_{\text{2D}}$ is set to its optimum value, found numerically by minimizing the total energy of $\phi_{\text{2D}}^d$. The corresponding binding energy is $E_b$. These quantities are shown as a function of nanoplatelet thickness in Suppl. Fig. 1a. Validation of this variational approach compared to a full numerical calculation may be found in Suppl. Note 1. Note that the hard-wall boundary conditions for the wavefunction in the $z$ direction do not take into account exciton leakage into the ligand layer. Leakage likely occurs in reality, so $E_b$ will be somewhat overestimated, $a_{\text{2D}}$ underestimated, and $\phi_{\text{2D}}^d(0)$ overestimated, resulting in lifetimes that are somewhat underestimated.

### Ligand fluctuation model

Here we calculate the function

$$A(\varepsilon) = \sum_i \left| \int d^2\mathbf{R}\psi_i(\mathbf{R}) \right|^2 \delta(\varepsilon - \varepsilon_i), \quad (9)$$

which is directly proportional to the absorption coefficient, and calculate the density of states $n(\varepsilon) = \sum_i \delta(\varepsilon - \varepsilon_i)$. We assume the potential $V(\mathbf{R})$ arising from nanoplatelet fluctuations is a "white-noise" potential, i.e., it obeys the correlation function

$$\langle V(\mathbf{R})V(\mathbf{R}')\rangle = \Gamma\delta(\mathbf{R} - \mathbf{R}'). \quad (10)$$

Such a potential was previously used to describe the fluctuation of the energy gap in alloy semiconductors[35,60]. The potential strength was found to be $\Gamma = \alpha^2 x(1 - x)/N$, where $x$ is the alloy content, $\alpha$ is the gap

change with alloy content, and $N$ is the density of alloy sites. We use the same expression for the ligand fluctuation model with $x$ now representing the ligand mixing fraction, $\alpha$ the gap change with ligand type, and $N$ the density of ligand sites, as discussed in the main text. We follow ref. 35 and introduce the broadening parameter $W$:

$$W = \frac{M\Gamma}{2\pi\hbar^2} = \alpha^2 \frac{x(1-x)}{N} \frac{M}{2\pi\hbar^2}, \tag{11}$$

which allows us to calculate the universal form of the exciton absorption line $A(\varepsilon/W)$ as a function of the dimensionless energy $\varepsilon^* = \varepsilon/W$. This dependence was originally derived for the three-dimensional case in ref. 34 and extended to the two-dimensional case in ref. 35.

At high energy $\varepsilon$ the exciton wavefunction is similar to the ideal case, and so the absorption lineshape function is given by perturbation theory[34,35]:

$$A_>(\varepsilon) = \frac{WS}{\varepsilon^2} + \mathcal{O}\left(\frac{1}{\varepsilon^3}\right). \tag{12}$$

At low energy we employ "optimal fluctuation theory" in which absorption is determined by the fluctuations that most strongly localize the exciton. The wavefunction $\psi$ subject to the optimal fluctuation is proportional to the test function $\varphi$ which is a stationary point of the functional

$$G[\varphi] = \int d^2r' \left( |\nabla\varphi|^2 + \varphi^2 - \frac{1}{2}\varphi^4 \right), \text{ where } r' = R/R_\varepsilon \text{ and } R_\varepsilon = \frac{\hbar}{\sqrt{2m|\varepsilon|}} \tag{13}$$

Using the ansatz $\varphi = Be^{-\eta r}$, we find

$$G[\varphi] = \frac{8B^2(1+\eta^2) - B^4}{32\eta^2}, \tag{14}$$

which has a saddle point at $B = 2\sqrt{2}, \eta = 1$. The normalized wavefunction is therefore

$$\psi_\varepsilon(R) = \sqrt{\frac{2}{\pi R_\varepsilon^2}} e^{-R/R_\varepsilon} \tag{15}$$

Therefore the low-energy limit of Eq. (9) is:

$$A_<(\varepsilon) = \left| \int d^2\mathbf{R}\psi_\varepsilon(\mathbf{R}) \right|^2 n(\varepsilon). \tag{16}$$

In this regime, the 2D density of states was found to be[61]

$$n_<(\varepsilon) \propto \frac{2m}{\hbar^2} \left( \frac{|\varepsilon|}{4\pi W} \right)^{0.569} \exp\left( -0.931 \frac{|\varepsilon|}{W} \right) \tag{17}$$

Ensuring the correct overall normalization $\int A(\varepsilon)d\varepsilon = S$, Eq. (16) then gives

$$A(\varepsilon) = 0.24S \frac{e^{-0.931|\varepsilon|/W}}{|\varepsilon|} \left( \frac{|\varepsilon|}{W} \right)^{0.569}. \tag{18}$$

Connecting these two forms following ref. 35 (illustrated in Suppl. Fig. 2a) and ensuring the correct overall normalization $\int A(\varepsilon)d\varepsilon = S$ is maintained, we arrive at

$$A(\varepsilon) = \frac{S}{W} A^*(\varepsilon^*) \tag{19}$$

$$A^*(\varepsilon^*) = \begin{cases} 0.24|\varepsilon^*|^{-0.43} e^{-0.93|\varepsilon^*|} & \text{if } \varepsilon^* < -1.5 \\ \frac{0.5}{(\varepsilon^*-0.67)^2 + 5.28} \left[ 1 + \left( 1 + \frac{0.86}{(\varepsilon^*+1.5)^2} \right)^{-1/2} \right] & \text{if } \varepsilon^* > -1.5. \end{cases} \tag{20}$$

In addition to $A(\varepsilon)$, we need the density of states $n(\varepsilon)$. At low energy, ref. 61 found that it is well approximated by the simple exponential

$$n(\varepsilon^*) = 0.17n_0 e^{0.931\varepsilon^*}. \tag{21}$$

At high energy, the density of states is well described by the coherent potential approximation[61] and is defined implicitly via

$$\varepsilon^* = \ln(\tilde{n}\csc(\tilde{n})) + 1 - \tilde{n}\cot(\tilde{n}), \text{ where } \tilde{n} = \frac{\pi n(\varepsilon^*)}{n_0}. \tag{22}$$

These may be connected smoothly, as illustrated in Fig. 2b.

$$n(\varepsilon^*) = n_{\text{low}}(\varepsilon^*)\frac{1}{2}\text{erfc}\left[3(\varepsilon^* - 0.596)\right] + n_{\text{high}}(\varepsilon^*)\frac{1}{2}\text{erfc}\left[-3(\varepsilon^* - 0.596)\right]. \tag{23}$$

The results of this procedure are dimensionless absorption lineshape $A^*(\varepsilon^*)$ and the dimensionless recombination rate $1/\tau^*(\varepsilon^*) = A^*(\varepsilon^*) n_0/n(\varepsilon^*)$. These functions apply to any semiconductor nanoplatelet subject to ligand disorder, and are plotted in Fig. 2.

### Confined fluctuations

We describe a fluctuation with energy $\varepsilon$ centered at $\mathbf{R}_f$ within a rectangular nanoplatelet (side lengths $L_x$ and $L_y$) as

$$\psi(\mathbf{R},\mathbf{R}_f,R_\varepsilon) = \mathcal{N}(\mathbf{R}_f,R_\varepsilon) e^{-|\mathbf{R}-\mathbf{R}_f|/R_\varepsilon} \cos\left(\frac{\pi X}{L_x}\right) \cos\left(\frac{\pi Y}{L_y}\right) \tag{24}$$

where $R_\varepsilon = \hbar/\sqrt{2M\varepsilon}$ and $\mathcal{N}(\mathbf{R}_f,R_\varepsilon)$ is defined such that

$$\int d^2\mathbf{R}|\psi(\mathbf{R},\mathbf{R}_f,R_\varepsilon)|^2 = 1 \tag{25}$$

Now to calculate $A(\varepsilon)$ for small $\varepsilon$, we average the fluctuation center $\mathbf{R}_f$ over the platelet:

$$A_<(\varepsilon) = \frac{n(\varepsilon)}{S} \int d^2\mathbf{R}_f \left| \int d^2\mathbf{R}\psi(\mathbf{R},\mathbf{R}_f,R_\varepsilon) \right|^2 \tag{26}$$

This is done numerically. We then match to the high-energy form:

$$A_>(\varepsilon) = \frac{S}{W} \frac{0.5}{(\varepsilon/W - a_3)^2 + a_1} \left[ 1 + \left( 1 + \frac{a_2}{((\varepsilon - \varepsilon_m)/W + 0.01)^2} \right)^{-1/2} \right]. \tag{27}$$

The matching point $\varepsilon_m$ is chosen to be the inflection point of $A_<(\varepsilon)$, and we fit $a_1$, $a_2$, and $a_3$ to match the value and derivative of $A$ at $\varepsilon_m$ and to satisfy the normalization condition $\int d\varepsilon A(\varepsilon) = S$. Note that $A_>$ fulfills Eq. (12) by construction. The resulting functions are shown in Fig. 2b.

### Density-functional theory

To calculate the shift of the gap upon passivation, CdSe nanoplatelets were simulated using density-functional theory (DFT). All DFT calculations used the VASP code with PAW pseudopotentials and a plane-wave energy cutoff of 500 eV.

Nanoplatelets were simulated using periodic boundary conditions with a single unit cell in the in-plane direction and between 2 and 7 monolayers in the out-of-plane direction, with periodic images separated by vacuum. An $N$-monolayer nanoplatelet is terminated by Cd on each surface, so it consists of $N$ Se layers and $N + 1$ Cd layers. This is sometimes referred to as $N.5$ monolayers. The thickness is defined between the centers of the surface atoms: $d = Na_c/2$. Atomic positions were relaxed using PBE, with van der Waals interactions taken into account with the DFT-D3 method of Grimme with zero damping (we call this PBE+vdW). The in-plane lattice constant of the nanoplatelets was fixed to the relaxed value of bulk zincblende CdSe using PBE+vdW, 0.613 nm.

Next, a bulk cell was relaxed using HSE+vdW, and the HSE mixing was tuned to match the zero-temperature gap of zincblende CdSe, 1.661 eV. A mixing parameter of 0.266 was found to yield a band gap of 1.662 eV and a lattice constant of 0.608 nm, in excellent agreement with the experiment. The same mixing parameter was used for the calculations under tensile strain, shown in Suppl. Fig. 11.

Finally, the nanoplatelets which had been relaxed in PBE+vdW were uniformly scaled to match the HSE+vdW lattice constant, and a static HSE+vdW calculation was performed on the scaled nanoplatelets. Suppl. Fig. 9 shows atomistic visualizations of the 4 ML nanoplatelets; nanoplatelets with other thicknesses are similar. Suppl. Note 3 discusses the identification of states as either surface states or confined bulk-like states.

### Radiative decay time

To describe the radiative decay time in nanoplatelets we will follow the approach developed in ref. [35]. The binding energy of the exciton in nanoplatelet is much larger than the exciton linewidth, and the 2D exciton is weakly confined in the $xy$ plane of the nanoplatelet and strongly confined in the out-of-plane $z$ direction, so its wavefunction can be written:

$$\psi(\mathbf{r}_e, \mathbf{r}_h) = u^c(\mathbf{r}_e)u^v(\mathbf{r}_h)\Psi(\mathbf{R})\phi^d_{2D}(\boldsymbol{\rho})\chi_e(z_e)\chi_h(z_h). \quad (28)$$

This is the same wavefunction written in Eq. (1), but now explicitly including $u^c$ and $u^v$, the Bloch functions of the conduction and valence bands respectively.

The radiative lifetime of a bright exciton state emitting light with frequency $\omega$ is given by

$$\frac{1}{\tau_{ex}} = \frac{4}{3}\frac{\omega n \alpha_f}{m_0^2 c^2}|\langle\psi|\hat{p}_\mu|\mathcal{G}\rangle|^2, \quad (29)$$

where $|\mathcal{G}\rangle$ is the vacuum state, $\hat{p}_\mu$ is the bright state's transition dipole, $n$ is refractive index of the media, $\alpha_f = 1/137$ is the fine structure constant, $m_0$ is the free electron mass, and c is the speed of light. The matrix element is

$$\langle\psi|\hat{p}_\mu|\mathcal{G}\rangle = P_{cv}\int d^3\mathbf{r}_e d^3\mathbf{r}_h \psi(\mathbf{r}_e, \mathbf{r}_h)\delta(\mathbf{r}_e - \mathbf{r}_h) = P_{cv}\phi^d_{2D}(0)\int d^2\mathbf{R}\,\psi(\mathbf{R}). \quad (30)$$

where $P_{cv}$ is the Kane matrix element (arising from the band-edge Bloch functions). The radiative decay rate can be written as[35]

$$\frac{1}{\tau_r(i)} = \frac{|\phi^d_{2D}(0)|^2 K_z}{\tau_0}\left|\int d^2R\psi_i(\mathbf{R})\right|^2, \quad (31)$$

where $K_z = |\int dz\chi_e(z)\chi_h(z)|^2 = 1$ and the characteristic lifetime $\tau_0$ is[35]

$$\frac{1}{\tau_0} = \frac{4}{3}\omega n \alpha_f \frac{P_{cv}^2}{m_0^2 c^2}. \quad (32)$$

The rate of exciton radiative recombination depends strongly on the exciton center of mass function, which is quantified by the factor $|\int d^2R\psi_i(\mathbf{R})|^2$. This factor has dimensions of area and represents the area of the coherent exciton center-of-mass motion. The radiative recombination described in Eq. (31) is enhanced by the ratio of this coherent area to the square of the exciton radius, which is proportional to $1/|\phi^d_{2D}(0)|^2$.

The exciton decay rate depends on the exciton state $i$, and it is reasonable to introduce the average decay rate at energy $\varepsilon$[34,35]:

$$\frac{1}{\tau_i(\varepsilon)} = \frac{\sum_i[1/\tau_r(i)]\delta(\varepsilon_i - \varepsilon)}{\sum_i\delta(\varepsilon_i - \varepsilon)} = \frac{|\phi^d_{2D}(0)|^2 K_z}{\tau_0}\frac{A(\varepsilon)}{n(\varepsilon)} \quad (33)$$

The population is spread across both the bright states [with density $n(\varepsilon)$] and dark states [with density $n(\varepsilon + \Delta E_{exch})$] according to a Boltzmann distribution, so Eq. (33) allows us to calculate the average decay rate of the exciton at temperature $T$:

$$\frac{1}{\tau_{avg}(T)} = \frac{|\phi^d_{2D}(0)|^2}{\tau_0}\frac{\int A(\varepsilon)\exp(-\varepsilon/k_B T)d\varepsilon}{\int[n(\varepsilon) + n(\varepsilon + \Delta E_{exch})]\exp(-\varepsilon/k_B T)d\varepsilon}. \quad (34)$$

We note that at low temperature the integrals in Eq. (34) as written diverge for large negative $\varepsilon$, since the population factor $\exp(-\varepsilon/k_B T)$ for the bosonic excitons grows faster than $n(\varepsilon)$ decays. To accurately describe the decay time in this regime, one must take into account the fact that the potential cannot reduce the energy of an exciton more than $-\alpha(1-x)$, corresponding to a completely passivated region of the platelet. We impose this physical constraint by applying a low-energy cutoff of the exciton density of states, setting $n(\varepsilon) = 0$ for $\varepsilon < -\alpha(1-x)$. With this cutoff applied, all the integrals converge.

## Data availability

The data from the DFT calculations of passivated CdSe nanoplatelets generated in this study are available in NOMAD repository https://doi.org/10.17172/NOMAD/2024.05.30-1. The data from the fluctuation model are provided in Supplementary Data 1. Source data are provided with this paper.

## Code availability

Python code implementing the ligand fluctuation model is provided in Supplementary Data 1.

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

## Acknowledgements

Thanks to Vladimir Lesnyak for the experimental data in Fig. 3b and to Todd D. Krauss, Jonathan Owen, Sandrine Ithurria, and Artsiom Antanovich for valuable discussion and useful comments on the manuscript. The authors acknowledge support by the Office of Naval Research through the Naval Research Laboratory's Basic Research Program.

## Author contributions

Al.L.E. and S.C.E. conceived the idea and supervised the project. M.W.S. and Al.L.E. built the theoretical model. M.W.S. performed the calculations and created the figures. All authors edited the figures and wrote and edited the paper.

## Competing interests

The authors declare no competing interests.
