## [Peer Review File · Nature Communications]

Controlling light emission from semiconductor nanoplatelets using surface chemistryEditorial Note: Parts of this Peer Review File have been redacted as indicated to maintain the confidentiality of unpublished data.

Reviewer #1 (Remarks to the Author):

Swift and co-workers report on the influence of a randomly fluctuating potential on the absorption/emission line width and lifetime of 2D excitons in weakly confined CdSe nanoplatelets. The source of the fluctuations is assigned to local variations of the ligand density, and calculations show that this results in a localization of the exciton center-of-mass wavefunction, and a significant line broadening. Their calculations highlight the importance of random potential fluctuations on the optical properties of colloidal 2D nanocrystals, and may find widespread interest and impact in the colloidal quantum dot community.

I do have some concerns on the comparison of their calculations with experimental data, which I believe that the authors should address.

Let me start by summarizing some optical properties, as obtained by experiment:

Regarding the fluorescence line width.

At room temperature, the CdSe emission line width (fwhm) varies as follows: 2ML, 63 meV; 3ML, 46 meV, 4ML, 44 meV, 5ML, 36 meV; 6ML, 35 meV; 7 ML, 38 meV. Data calculated from 10.1021/acs.chemmater.0c03066 and 10.1021/acs.nanolett.8b02361, and in line with ref 2, 5 and ref 31. They show that the line width decreases for increasing nanoplatelet thickness.

At cryogenic temperature, regardless of the thickness, the line width is substantially narrower: 3 ML, 20 meV; 4 ML, 17 meV, 5 ML, 15 meV. Data taken from ref 5, which are in line with ref 16 and 10.1039/d0nr04745g for 4 ML nanoplatelets.

Single nanoplatelets show an even smaller line width at cryogenic temperature, on the order of 1 meV. Data taken from ref 32, in line with 10.1021/nl301071n and 10.1021/acs.nanolett.9b02856.

Regarding the emission lifetime.

At room temperature, values vary between 2.5 ns (2ML) and 11 ns (7ML), 10.1021/acs.chemmater.0c03066 and 10.1021/acs.nanolett.8b02361. Thinner nanoplatelets typically show faster lifetimes.

At cryogenic temperature, again irrespective of thickness, values decrease to about 15-125 ps (10.1039/D0NR03170D), 100 ps (ref 5), 150 ps (ref 32) or 340 ps (ref 2). Ensembles (10.1039/D0NR03170D, ref 5) and single particles (ref 32) show similar results.

Regarding the ligand density.

A ligand density of about 5.3 ligands / nm² is reported experimentally (ref 29).

This brings me to a few points:

1. First, as a general comment, when reporting on the experimental values throughout the manuscript, please always include the temperature for which they are valid, since both the line width and the lifetime are temperature-dependent.

2. For some statements, I come to a different conclusion:

“second, because the linewidths from individual nanoplatelets are very similar to those of

ensembles”.

“However, the emission has a unexpectedly large linewidth (in the range 35–55 meV) that remains unexplained. As mentioned above, even single nanoplatelets show these linewidths and thus we can rule out variations in their lateral size as a broadening mechanism.”

I agree that at room temperature, photoluminescence excitation spectra and comparison of single-particle with ensemble fluorescence shows no heterogeneous broadening. However, as discussed above, the typical line width for single platelets is on the order of 1 meV at cryogenic temperature, substantially smaller than 35-55 meV.

3. This brings me to the influence of temperature on the line width. It seems that this is not included in the manuscript. Can one assume that the ligand density and its impact on the line width could be considered temperature-independent, and wouldn't the calculations then not predict a temperature-independent line width, in contrast with observation? While agreement is obtained between calculations and room-temperature data, the fact that smaller line widths are measured at cryogenic temperature should be discussed in the manuscript. Possibly, the authors are overestimating the broadening, it would be instructive to compare calculations to the experimental line widths obtained at cryogenic temperature, and see what ligand deficiency 'x' is needed to achieve agreement with experiments.

4. This also means that phonon-coupling is still a viable source of line broadening at higher temperature. I would briefly address this in the manuscript.

5. When including the updated discussion on the ligand density, I believe that it's also worth mentioning that ref 29 actually obtained an experimental ligand density that is quite close to the calculated maximum. Hence, while the influence of the random potential is unambiguous, perhaps other sources beyond ligand variations might be behind it, and one could perhaps suggest what form these might take.

6. On the fluorescence lifetime, calculations appear quite in line with experimental lifetimes at cryogenic temperature, obtained under nonresonant excitation (see data above and calculations in figure 4). However, calculated values at room temperature are more than a magnitude faster than experimental results. Please add a brief discussion on this discrepancy.

7. On the assignment of the low-energy emission to a dark state. In ref 5 and ref 8, the optical properties of this low-energy state were measured. They showed that this state also shows no significant dependence on magnetic field and no significant degree of circular polarization. Are these observations in line with the current observation of LA-assisted emission from a dark state? Please discuss.

8. As a smaller comment: can the authors report on the typical potential fluctuations (in eV/meV) in the manuscript? This would help to appreciate what values are required to obtain the associated exciton center-of-mass localization. It could perhaps accompany figure S6.

9. As a smaller comment, I find this statement somewhat brief: “the average exciton lifetime increases with increasing temperature as more of the slow states in the high-energy tail become populated.” The fact that higher-energy states are actually slower may benefit from a short clarification.

Reviewer #2 (Remarks to the Author):

Swift et al show that one plausible explanation for the broadening of the photoluminescence lines in atomically-precise nanoplatelets is that the structures have heterogeneous ligand coverage which induces variable confinement potential within single nanoplatelets. I am not as optimistic as the authors maybe that a complete ligand shell is possible, but I think this work represents an important development for conceptual understanding of the origin of line-broadening distinct from thermal broadening and size-dispersion. It is (presumably) operative in other nanocrystal systems too, although not as important. I find the line of argument very reasonable and enlightening (even if the specific mechanism is more complicated than described) and the manuscript should be published with minor changes. My largest suggestions are to provide some indication about the relationship between the proposed variability in ligand coverage as a basis for the broadening of PL lines and alternatives which could have a related origin. In particular, I would note that several works have demonstrated that the lattice of the NPLs themselves depends on the ligand coating (e.g. DOIs: 10.1039/C7NR05065H, 10.1021/acsnano.8b09794, 10.1021/acs.chemmater.0c02305), although I do not know of any case in which the ligand binding fraction is convincingly measured and correlated with properties. In this case, it would at least seem plausible that ligand coverage is changing the potential at different parts of the well via modification of the lattice unit cells. A related “mechanism” could also be the mechanical deformation of the NPLs (in practice they are not flat sheets), which also should generate a dispersion in the unit cells and therefore broadening the PL. Are there experimental ways to distinguish between these possibilities?

Minor:

-In support of the authors' arguments, I would note that mercury telluride and mercury selenide systems which are synthesized with mixtures of ligands (amine/ammonium, oleate) have substantial broadening of the absorption/emission bands. (See e.g. DOIs: 10.1002/adom.202302004,)

-The discussion in the text and presentation of Figure 2 was challenging to follow.

-How does the data on broadening relate to existing photoluminescence excitation experiments? This seems to offer a potential challenge: most demonstrations show that PLE data is superimposable on both sides of the PL peak. If there are indeed parts of the NPL emitting at longer wavelengths, should we expect to see funneling of excitations to the lowest energy emitting regions? Or, do the authors consider the well depth sufficiently small that this does not happen at RT? Or possibly, the changes are fluxional?

Reviewer #3 (Remarks to the Author):

- This theoretical paper claims that empty surface sites of CdSe nanoplatelets can explain several optical features of these objects such as larger than expected emission lines. As far as i can judge the calculations presented are sound. However i have concerns about the soundness of the hypotheses with regard to our current experimental knowledge on the surface chemistry of NPL which prevents me from recommending acceptance to Nature Communications.

- My main concern is about the possibility that a significant fraction of the surface sites of CdSe NPL (i.e. Cd²⁺ ions) are not linked to carboxylates. The paper argues that this is this incomplete coverage is the source a spatially fluctuating potential leading to optical broadening at the individual NPL level. The authors justify this assumption by stating that "full occupancy of every bonding site on the NPL is statistically very unlikely due to the steric interference". However, i dont think this is a realistic assumption. Singh et al (ref 29) showed that the density of ligands at the surface totally compensate the surface charges with a density of carboxylate groups close to the one of Cd²⁺ at the surface (quote of ref 29: A ligand surface concentration of 5.3 nm⁻² closely matches the surface concentration of 5.4 nm⁻² of Cd²⁺ cations at the CdSe (100) surface, indicating that the single excess monolayer of Cd²⁺ is charge compensated by two layers of carboxylates.). Linewidths between 35 and 55 meV as the one observed in experiments are predicted in the framework of the paper for ligand coverages of 50%. Such a large fraction of empty sites is far beyond the precision of the measurement and would have been detected experimentally which is not the case. Another argument against this assumption is that such a large fraction of Cd²⁺ atoms at the surface would yield highly charged NPL which are very unlikely to be stable in apolar organic solvent. Half Cd²⁺ empty sites would yield a surface charge density of 10 C.nm⁻² which is an order of magnitude larger than the surface charge density of fully ionized polar surfaces in water. How these charges are supposed to be compensated in organic solvents where there is no counter-ions ? Such a high charge surface density would also be detected by zeta potential measurements which is at odds with all the litterature reports i have in mind. Overall, the consensus in the colloidal nanocrystal community is that even if defects can exist, the surface is fully covered by ligand's fonctionnal group. I would be happy to update my prior on this but the argument presented here as quoted is not convincing at all since steric interference is perfectly compatible with a full coverage monolayer. The packing density of crystalline alkyl chains is 4.9 nm⁻² i.e. very close to the surface density of Cd atoms. Since mixtures of long and short chains carboxylate are known to be both present at NPL surface, this gives even more room for the ligands' chains.

- I can see at least two other phenomena would could explain the line broadening. First, it is likely that the surface of NPL is not homogeneous since both chains ligands (acetate) and long chain (oleate or myristate) ligands are necessary to synthesize NPL. Hence, there might be patches of acetate and oleate at the surfaces. This would certainly yield variations in the dielectric permittivity along the NPL surface but in this case there is no charge which is way more realistic. Could the authors comment on this hypothesis ? Would such differences in dielectric permittivity be able to contribute to the phenomenon which the authors try to explain ?

- It is also possible that surface stress imposed by the ligands induce inhomogeneities in the strain in a single NPL. For example, the edges could be more or less strained than the center. This could also be a plausible source of the effect observed with variations of the band gap in a unique NPL. Could the authors comment on this ?

- Finally, this surface stress is known to induce a significant curvature to the NPL. In all the electronic calculations, the NPL is supposed to be flat. I understand that all the tools derived from the bulk electronic structure calculation are impossible to use for curved NPL because it forbids periodic boundary conditions but how could curvature impact the band gap ?

- A previous paper on DFT calculations of CdSe NPL band gaps (10.1021/acs.jpcc.0c10559)

concludes that "it is critical to account for surface stress effects and consider a finite, rather than infinite, potential barrier when describing quantum confinement". I don't think surface stress has been taken into account in this study nor finite potential barrier. How could that impact the findings presented ?

- The authors claim that the predicted variation of the radiative lifetime with temperature (figure 4d) is in qualitative agreement with experiments. However I can't find a plot presenting lifetime measurements as a function of temperature in the two references cited.

- The title is too broad and does not convey the main claim of the paper.

Dear Editor and Reviewers,

Thank you for the detailed and insightful comments. After significantly re-writing the manuscript, modifying some central assumptions about the existence of unpassivated surface sites, and including a substantial amount of new modeling work, we believe that we have successfully addressed the reviewer comments. We are therefore resubmitting our manuscript to *Nature Communications*.

Please find the reviewer comments reproduced below in blue, along with our point-by-point responses. A marked version of the manuscript showing our changes is included with our resubmission. Note that all the figures in the main text and many Supplementary figures have also been updated based on our new results.

Reviewer #1 (Remarks to the Author):

Swift and co-workers report on the influence of a randomly fluctuating potential on the absorption/emission line width and lifetime of 2D excitons in weakly confined CdSe nanoplatelets. The source of the fluctuations is assigned to local variations of the ligand density, and calculations show that this results in a localization of the exciton center-of-mass wavefunction, and a significant line broadening. Their calculations highlight the importance of random potential fluctuations on the optical properties of colloidal 2D nanocrystals, and may find widespread interest and impact in the colloidal quantum dot community.

We thank the reviewer for their positive assessment of the work and its impact.

I do have some concerns on the comparison of their calculations with experimental data, which I believe that the authors should address.

Let me start by summarizing some optical properties, as obtained by experiment:

Regarding the fluorescence line width.

At room temperature, the CdSe emission line width (fwhm) varies as follows: 2ML, 63meV; 3ML, 46 meV, 4ML, 44 meV, 5ML, 36 meV; 6ML, 35 meV; 7 ML, 38 meV. Data calculated from 10.1021/acs.chemmater.0c03066 and 10.1021/acs.nanolett.8b02361, and in line with ref 2, 5 and ref 31. They show that the line width decreases for increasing nanoplatelet thickness.

At cryogenic temperature, regardless of the thickness, the line width is substantially narrower: 3 ML, 20 meV; 4 ML, 17 meV, 5 ML, 15 meV. Data taken from ref 5, which are in line with ref 16 and 10.1039/d0nr04745g for 4 ML nanoplatelets.

Single nanoplatelets show an even smaller line width at cryogenic temperature, on the order of 1 meV. Data taken from ref 32, in line with 10.1021/nl301071n and 10.1021/acs.nanolett.9b02856.

Regarding the emission lifetime.

At room temperature, values vary between 2.5 ns (2ML) and 11 ns (7ML), 10.1021/acs.chemmater.0c03066 and 10.1021/acs.nanolett.8b02361. Thinner nanoplatelets typically show faster lifetimes.

At cryogenic temperature, again irrespective of thickness, values decrease to about 15-125 ps (10.1039/D0NR03170D), 100 ps (ref 5), 150 ps (ref 32) or 340 ps (ref 2). Ensembles (10.1039/D0NR03170D, ref 5) and single particles (ref 32) show similar results.

Regarding the ligand density.

A ligand density of about 5.3 ligands / nm² is reported experimentally (ref 29).

Thanks to the reviewer for this very useful summary of measured optical properties. We have added the suggested citations that were missing from our original manuscript, and expanded the experimental comparisons throughout. The results summarized here also inform our answers to the points below.

This brings me to a few points:

1. First, as a general comment, when reporting on the experimental values throughout the manuscript, please always include the temperature for which they are valid, since both the line width and the lifetime are temperature-dependent.

We have been careful to do so throughout the revised manuscript. In particular, Figure 4b has direct experimental comparisons of the linewidth at varying temperatures, taking the effect of thermal broadening into account. Discussion of the comparison with the temperature-dependent experimental lifetimes has also been expanded (lines 252-260; Supplementary Section S-VI).

2. For some statements, I come to a different conclusion:

“second, because the linewidths from individual nanoplatelets are very similar to those of ensembles”. “However, the emission has a unexpectedly large linewidth (in the range 35–55 meV) that remains unexplained. As mentioned above, even single nanoplatelets show these linewidths and thus we can rule out variations in their lateral size as a broadening mechanism.”

I agree that at room temperature, photoluminescence excitation spectra and comparison of single-particle with ensemble fluorescence shows no heterogeneous broadening. However, as discussed above, the typical line width for single platelets is on the order of 1 meV at cryogenic temperature, substantially smaller than 35-55 meV.

This gets to the important distinction between emission and absorption linewidths. Though emission linewidths are the practical motivation, the model in fact describes absorption linewidths. Absorption is simpler because it is independent of temperature, and emission and absorption linewidths should be similar at room temperature. In principle, one could simulate the temperature-dependent emission lineshape by convolving the absorption lineshape with a thermal occupation function. However, this neglects important physics: for example, at cryogenic temperatures, the meV-width emission may be better described as emission of nearly indistinguishable single photons from the lowest exciton state. Therefore, we have elected to remain focused on absorption linewidths. We have added some important clarifying discussion of the low-temperature single-photon emission and emission vs absorption lineshapes (lines 105-113).

3. This brings me to the influence of temperature on the line width. It seems that this is not included in the manuscript. Can one assume that the ligand density and its impact on the line width could be considered temperature-independent, and wouldn't the calculations then not predict a temperature-independent line width, in contrast with observation? While agreement is obtained between calculations and room-temperature data, the fact that smaller line widths are measured at cryogenic temperature

should be discussed in the manuscript. Possibly, the authors are overestimating the broadening, it would be instructive to compare calculations to the experimental line widths obtained at cryogenic temperature, and see what ligand deficiency 'x' is needed to achieve agreement with experiments.

Indeed there is temperature-dependent broadening in experiments. In the revised manuscript, we take this into account phenomenologically, as a simple broadening of the line by $k_B T$ (Figure 1c & caption; Figure 4b & caption). As you point out, the ligand-fluctuation-induced broadening should be temperature independent. Our comparison to experiment in Figure 4b now uses cryogenic absorption linewidths (orange) and room-temperature linewidths with thermal broadening subtracted off (green). The linewidths are smaller in the revised manuscript (because, based on feedback from reviewer 3 we focus on disorder between types of ligands rather than ligand occupancy). Agreement with experimental linewidths is now fairly good.

4. This also means that phonon-coupling is still a viable source of line broadening at higher temperature. I would briefly address this in the manuscript.

We have added discussion of this point (lines 212-219). This also affects the lifetimes, as we point out subsequently.

5. When including the updated discussion on the ligand density, I believe that it's also worth mentioning that ref 29 actually obtained an experimental ligand density that is quite close to the calculated maximum. Hence, while the influence of the random potential is unambiguous, perhaps other sources beyond ligand variations might be behind it, and one could perhaps suggest what form these might take.

Based on this result, the feedback of Reviewer 3, and extended conversations with a number of experimental colleagues, we have become convinced that unpassivated sites are in fact quite rare. We now believe that disorder between types of ligands, especially short- and long-chain ligands, is the most likely source of broadening (lines 52-56). We use disorder between types of small ligands as a simple proxy, given the computational challenge of addressing large ligand layers (lines 59-67).

We have also emphasized that the general model is independent of the nature of the fluctuations (lines 49-51), and listed a few more possible sources of fluctuations that could be investigated in the future using the model if appropriate α values can be calculated. (lines 56-58)

6. On the fluorescence lifetime, calculations appear quite in line with experimental lifetimes at cryogenic temperature, obtained under nonresonant excitation (see data above and calculations in figure 4). However, calculated values at room temperature are more than a magnitude faster than experimental results. Please add a brief discussion on this discrepancy.

As part of the revision, we found smaller values of α , and therefore smaller linewidths, compared to the old results using occupancy disorder. This required some additional modeling: the effect of lateral confinement within the nanoplatelet on the fluctuations is now taken into account (Methods Section C: lines 335-344). This also acts to increase the lifetimes, particularly at room temperature, partially addressing this concern. However, the main impact likely comes from phonon-induced decoherence at higher temperature. We have added a brief discussion of this point (lines 252-260) and included a phenomenological model at the end of the SI (Section S-VI) which shows that a rough model of phonon-induced decoherence brings predicted lifetimes into the same range as experimental values.

7. On the assignment of the low-energy emission to a dark state. In ref 5 and ref 8, the optical properties of this low-energy state were measured. They showed that this state also shows no significant dependence on magnetic field and no significant degree of circular polarization. Are these observations in line with the current observation of LA-assisted emission from a dark state? Please discuss.

[Redacted]

8. As a smaller comment: can the authors report on the typical potential fluctuations (in eV/meV) in the manuscript? This would help to appreciate what values are required to obtain the associated exciton center-of-mass localization. It could perhaps accompany figure S6.

This is an excellent suggestion, which we discuss (lines 204-206) and have implemented in Figure S6b .

9. As a smaller comment, I find this statement somewhat brief: “the average exciton lifetime increases with increasing temperature as more of the slow states in the high-energy tail become populated.” The fact that higher-energy states are actually slower may benefit from a short clarification.

We have added a brief clarification and an appropriate citation (lines 169-171)

Reviewer #2 (Remarks to the Author):

Swift et al show that one plausible explanation for the broadening of the photoluminescence lines in atomically-precise nanoplatelets is that the structures have heterogeneous ligand coverage which induces variable confinement potential within single nanoplatelets. I am not as optimistic as the authors maybe that a complete ligand shell is possible, but I think this work represents an important development for conceptual understanding of the origin of line-broadening distinct from thermal broadening and size-dispersion. It is (presumably) operative in other nanocrystal systems too, although not as important. I find the line of argument very reasonable and enlightening (even if the specific mechanism is more complicated than described) and the manuscript should be published with minor changes.

We thank the reviewer for expressing the importance and broad applicability of the work.

My largest suggestions are to provide some indication about the relationship between the proposed variability in ligand coverage as a basis for the broadening of PL lines and alternatives which could have a

related origin. In particular, I would note that several works have demonstrated that the lattice of the NPLs themselves depends on the ligand coating (e.g. DOIs: 10.1039/C7NR05065H, 10.1021/acsnano.8b09794, 10.1021/acs.chemmater.0c02305), although I do not know of any case in which the ligand binding fraction is convincingly measured and correlated with properties. In this case, it would at least seem plausible that ligand coverage is changing the potential at different parts of the well via modification of the lattice unit cells. A related “mechanism” could also be the mechanical deformation of the NPLs (in practice they are not flat sheets), which also should generate a dispersion in the unit cells and therefore broadening the PL. Are there experimental ways to distinguish between these possibilities?

We have added discussion of additional mechanisms and included these suggested references (lines 52-58). We hope that our suggested experiments, correlating uniformity of ligand coverage with linewidths, will support our proposed broadening mechanism. If experiments can modulate the mechanical deformation and observe any resulting changes in linewidth, this would also be a valuable contribution.

Minor:

-In support of the authors' arguments, I would note that mercury telluride and mercury selenide systems which are synthesized with mixtures of ligands (amine/ammonium, oleate) have substantial broadening of the absorption/emission bands. (See e.g. DOIs: 10.1002/adom.202302004,)

We thank the reviewer for bringing this reference to our attention; we have included it in the revised manuscript (lines 20-21). This shows clearly that the phenomena we are modeling extend well beyond CdSe.

-The discussion in the text and presentation of Figure 2 was challenging to follow.

We have attempted to make this easier to follow. In particular, we have added some more explanation of the dimensionless units (lines 146-151).

-How does the data on broadening relate to existing photoluminescence excitation experiments? This seems to offer a potential challenge: most demonstrations show that PLE data is superimposable on both sides of the PL peak. If there are indeed parts of the NPL emitting at longer wavelengths, should we expect to see funneling of excitations to the lowest energy emitting regions? Or, do the authors consider the well depth sufficiently small that this does not happen at RT? Or possibly, the changes are fluxional?

By “superimposability”, we assume the author is referring to the fact that PLE lineshapes are nearly identical for fixed emission energies, both above and below the PL peak (as in Figure 4 of 10.1021/acs.nanolett.8b02361, now Ref 18). This phenomenon suggests that exciton population thermalizes at room temperature faster than the recombination time. The fluctuations are not deep enough to trap all excitons: instead, they add a low-energy tail to the density of states and broaden the emission line on average.

Reviewer #3 (Remarks to the Author):

- This theoretical paper claims that empty surface sites of CdSe nanoplatelets can explain several optical features of these objects such as larger than expected emission lines. As far as I can judge the calculations presented are sound. However I have concerns about the soundness of the hypotheses with

regard to our current experimental knowledge on the surface chemistry of NPL which prevents me from recommending acceptance to Nature Communications.

We thank the reviewer for their positive assessment of the calculations. We have taken their concerns regarding surface chemistry to heart and made substantial revisions. We believe the new manuscript now paints a more accurate picture of the surface chemistry.

- My main concern is about the possibility that a significant fraction of the surface sites of CdSe NPL (i.e. Cd²⁺ ions) are not linked to carboxylates. The paper argues that this incomplete coverage is the source of a spatially fluctuating potential leading to optical broadening at the individual NPL level. The authors justify this assumption by stating that "full occupancy of every bonding site on the NPL is statistically very unlikely due to the steric interference". However, I don't think this is a realistic assumption. Singh et al (ref 29) showed that the density of ligands at the surface totally compensates the surface charges with a density of carboxylate groups close to the one of Cd²⁺ at the surface (quote of ref 29: A ligand surface concentration of 5.3 nm⁻² closely matches the surface concentration of 5.4 nm⁻² of Cd²⁺ cations at the CdSe (100) surface, indicating that the single excess monolayer of Cd²⁺ is charge compensated by two layers of carboxylates.). Linewidths between 35 and 55 meV as the one observed in experiments are predicted in the framework of the paper for ligand coverages of 50%. Such a large fraction of empty sites is far beyond the precision of the measurement and would have been detected experimentally which is not the case. Another argument against this assumption is that such a large fraction of Cd²⁺ atoms at the surface would yield highly charged NPL which are very unlikely to be stable in apolar organic solvent. Half Cd²⁺ empty sites would yield a surface charge density of 10 C.nm⁻² which is an order of magnitude larger than the surface charge density of fully ionized polar surfaces in water. How these charges are supposed to be compensated in organic solvents where there is no counter-ions? Such a high charge surface density would also be detected by zeta potential measurements which is at odds with all the literature reports I have in mind. Overall, the consensus in the colloidal nanocrystal community is that even if defects can exist, the surface is fully covered by ligand's functional group. I would be happy to update my prior on this but the argument presented here as quoted is not convincing at all since steric interference is perfectly compatible with a full coverage monolayer. The packing density of crystalline alkyl chains is 4.9 nm⁻² i.e. very close to the surface density of Cd atoms. Since mixtures of long and short chain carboxylates are known to be both present at NPL surface, this gives even more room for the ligands' chains.

Thanks to the reviewer for pointing out the unrealistic surface chemistry in the previous manuscript. We took a closer look at Singh et al, and find their conclusions compelling. We have eliminated unpassivated surfaces from the revised manuscript, since we now believe they are unlikely to occur. Instead, we argue that disorder between different types of ligands, such as short- and long-chain carboxylates, is the most likely source of the fluctuations (lines 52-67).

We agree that a nanoplatelet with half of the surface Cd as bare Cd²⁺ would be highly charged and is extremely unrealistic. This was not the situation modeled in the previous version. Our modeled unpassivated slabs were electrically neutral, so the unpassivated sites are better described as Cd⁺, i.e., Cd²⁺ with a dangling bond state occupied by an electron. While not as clearly unphysical as a nanoplatelet with a large positive charge, we have realized that such a situation is also unrealistic. Such a dangling bond state would be highly reducing and would likely react with a ligand, a solvent molecule, or another solvated species, and subsequently the site would end up passivated in some way. We therefore assume all sites are passivated.

- I can see at least two other phenomena would could explain the line broadening. First, it is likely that the surface of NPL is not homogeneous since both chains ligands (acetate) and long chain (oleate or myristate) ligands are necessary to synthesize NPL. Hence, there might be patches of acetate and oleate at the surfaces. This would certainly yield variations in the dielectric permittivity along the NPL surface but in this case there is no charge which is way more realistic. Could the authors comment on this hypothesis ? Would such differences in dielectric permittivity be able to contribute to the phenomenon which the authors try to explain ?

The dielectric contrast between the ligands and the solvent is likely not sufficient for short- and long-chain ligand fluctuations to exert a large influence via the dielectric permittivity. However, this disorder is an important source of strain fluctuations, which couple to the gap. We now believe this is one of the most likely sources of the fluctuations in CdSe nanoplatelets, as discussed in the manuscript (lines 52-67). A detailed investigation is computationally challenging and beyond the scope of this paper, so for now we use a simpler DFT proxy, but a comprehensive computational study of chain-length disorder will definitely be of interest in the future.

- It is also possible that surface stress imposed by the ligands induce inhomogeneities in the strain in a single NPL. For example, the edges could be more or less strained than the center. This could also be a plausible source of the effect observed with variations of the band gap in a unique NPL. Could the authors comment on this ?

This is also plausible; we have added a discussion (lines 56-58).

- Finally, this surface stress is known to induce a significant curvature to the NPL. In all the electronic calculations, the NPL is supposed to be flat. I understand that all the tools derived from the bulk electronic structure calculation are impossible to use for curved NPL because it forbids periodic boundary conditions but how could curvature impact the band gap ?

Curvature would act primarily by applying different strain to the outside and inside of the curved sheet. This would modify the in-plane confinement somewhat, since strained regions will have a smaller gap than unstrained regions. Increased curling near the edges could also lead to broadening. We have mentioned these possibilities (lines 56-58).

- A previous paper on DFT calculations of CdSe NPL band gaps ([10.1021/acs.jpcc.0c10559](https://doi.org/10.1021/acs.jpcc.0c10559)) concludes that "it is critical to account for surface stress effects and consider a finite, rather than infinite, potential barrier when describing quantum confinement". I dont think surface stress has been taken into account in this study nor finite potential barrier. How could that impact the findings presented ?

Surface stress was partially taken into account by allowing the thickness of the platelet to relax during the DFT simulation, though full treatment of the stress effects of the ligands is beyond the scope of the paper. We would like to emphasize that we do not use the DFT gaps themselves for the photon emission/absorption energies, instead relying on experiment (see Figure S5). We only look at differences of DFT gaps between different passivating ligands, so inaccuracies in the DFT gap should mostly cancel out. We have added a brief discussion to this effect (lines 192-196).

Our use of hard-wall boundary conditions for the calculation of the quasi-2D exciton binding energy mean that the exciton energy is likely somewhat overestimated and the radius underestimated. As a

result, $\phi_{2D}^d(0)$ is overestimated, resulting in lifetimes that are somewhat underestimated. We have discussed this limitation in the Methods section (lines 297-300).

- The authors claim that the predicted variation of the radiative lifetime with temperature (figure 4d) is in qualitative agreement with experiments. However i cant find a plot presenting lifetime measurements as a function of temperature in the two references cited.

We have added more detailed lifetime comparisons, including a few new references (lines 252-260). We also grapple with the fact that, although lifetimes at cryogenic temperatures are described reasonably well, our model predicts significantly shorter room-temperature lifetimes than are observed experimentally. One factor is the inclusion of the effects of lateral confinement within the nanoplatelet on the fluctuations (Methods Section C: lines 335-344), which increases the lifetimes. We also discuss the possibility of phonon-induced decoherence at high temperature. A rough phenomenological model (SI Section S-VI) shows that this can explain the discrepancy with experiment.

- The title is too broad and does convey the main claim of the paper.

We believe the title is appropriately broad given the broad applicability of the fluctuation model. We have attempted to justify this by emphasizing that the model is not only applicable to the specific source of disorder we discuss, but to any disorder that can be parametrized by a gap change α (lines 49-58).

Reviewer #1 (Remarks to the Author):

The authors have made a substantial effort to address earlier comments, with additional calculations and associated discussion of the results. I have no further remarks.

Reviewer #2 (Remarks to the Author):

I am satisfied with the revisions made in the manuscript and support publication in its current form.

Reviewer #3 (Remarks to the Author):

Swift et al. have revised their paper significantly, according to the referees' reports. I would like to thank the authors for their modifications. Most notably, they modified a key aspect of the first version that was not physically sound.

However, I am still concerned that the paper makes strong claims not properly evidenced by their model. In short, I don't think the experimentally observed emission properties of nanoplatelets can be explained by the model exposed in the paper. In their rebuttal, they write, "We argue that disorder between different types of ligands such as short and long chain carboxylates is the most likely source of the fluctuations". However, this is not what is calculated in the paper. Instead, they show that different types of ligand *binding groups* can be the source of the fluctuations. As far as I understand the paper, the physical explanation for the fluctuation is that different binding groups, such as carboxylates, thiolates, iodine, and chlorine, induce local band gap fluctuations when mixed at the nanoplatelet's surface. I find the rationale perfectly sound and trust the authors' calculations are correct. However, in all the experimental cited studies on the optical properties of NPL, the ligand monolayer is exclusively composed of carboxylates (acetate and oleate/myristate); hence, the inhomogeneity in the ligand layer is *not in the binding group* but in the length of the alkyl chain of the ligand. Hence, the large experimental linewidth can not be explained by inhomogeneities in binding groups because there is none in the experiments. In this regard, the paper is misleading. For example, figure 1 shows "real nanoplatelets" with mixtures of ligands having different ligand chain lengths and different binding groups, but the paper only considers different binding groups. In the same line, the abstract claims that "the linewidth is controlled by inhomogeneities in the ligand layer," but in fact, the paper only proves this is true for inhomogeneities in binding groups. This is interesting but does not resolve the "linewidth puzzle," as claimed in the paper, because there are no inhomogeneities in binding groups in NPL that display this "linewidth puzzle."

Unless the authors can convincingly explain why different chain lengths could lead to strong variations in the band gap, the paper's abstract, introduction, and conclusion should be significantly modified to tone down the strong claims made, which are not backed up by sufficient evidence. I think the model is very interesting and worth publishing on its merit, but in the current version of the paper, the broad impact of solving unexplained experimental findings is oversold.

Reviewer #3 (Remarks on code availability):

There is a readme file that details the different files of the code.

I ran the python notebook flawlessly on my machine but did not review the code further.

Dear Editor and Reviewers,

Thank you for carefully reviewing our manuscript, and for the opportunity to make additional revisions. As requested by Reviewer 3, we have added additional evidence to support our claims of solving an experimental puzzle. Their comments are reproduced below, along with our response.

Reviewer #3 (Remarks to the Author):

Swift et al. have revised their paper significantly, according to the referees' reports. I would like to thank the authors for their modifications. Most notably, they modified a key aspect of the first version that was not physically sound.

We thank the Reviewer again for pointing out this issue in our original manuscript.

However, I am still concerned that the paper makes strong claims not properly evidenced by their model. In short, I don't think the experimentally observed emission properties of nanoplatelets can be explained by the model exposed in the paper. In their rebuttal, they write, "We argue that disorder between different types of ligands such as short and long chain carboxylates is the most likely source of the fluctuations". However, this is not what is calculated in the paper. Instead, they show that different types of ligand *binding groups* can be the source of the fluctuations. As far as I understand the paper, the physical explanation for the fluctuation is that different binding groups, such as carboxylates, thiolates, iodine, and chlorine, induce local band gap fluctuations when mixed at the nanoplatelet's surface. I find the rationale perfectly sound and trust the authors' calculations are correct. However, in all the experimental cited studies on the optical properties of NPL, the ligand monolayer is exclusively composed of carboxylates (acetate and oleate/myristate); hence, the inhomogeneity in the ligand layer is *not in the binding group* but in the length of the alkyl chain of the ligand. Hence, the large experimental linewidth can not be explained by inhomogeneities in binding groups because there is none in the experiments. In this regard, the paper is misleading. For example, figure 1 shows "real nanoplatelets" with mixtures of ligands having different ligand chain lengths and different binding groups, but the paper only considers different binding groups. In the same line, the abstract claims that "the linewidth is controlled by inhomogeneities in the ligand layer," but in fact, the paper only proves this is true for inhomogeneities in binding groups. This is interesting but does not resolve the "linewidth puzzle," as claimed in the paper, because there are no inhomogeneities in binding groups in NPL that display this "linewidth puzzle."

Unless the authors can convincingly explain why different chain lengths could lead to strong variations in the band gap, the paper's abstract, introduction, and conclusion should be significantly modified to tone down the strong claims made, which are not backed up by sufficient evidence. I think the model is very interesting and worth publishing on its merit, but in the current version of the paper, the broad impact of solving unexplained experimental findings is oversold.

Indeed, we have used ligand type disorder (which we can calculate) as a proxy for ligand chain-length disorder (which is quite difficult to calculate fully). The reviewer's point is well taken: these two types of disorder interact with the exciton in different ways, so the appropriateness of the proxy is not obvious. We thank the reviewer for pointing out this weak point in our argument and welcome the opportunity to strengthen it. Our manuscript has been much improved by including an explanation of how chain-length disorder can lead to broadening via lattice strain, which shows that the effect has a comparable magnitude to that of ligand-type disorder. This discussion may be found on lines 205-238 and in Supplementary Figure S11. We have also addressed the referee's concern that this aspect of the paper could be misleading by adding some additional clarifying text in lines 68-70 to make sure that our proxy procedure is clear.

Ref 34 showed that long-chain carboxylate ligands (specifically oleate) induce tensile strain on the CdSe lattice. Our calculations indicate that short-chain carboxylates (specifically acetate) induce negligible strain. Modulation of semiconductor band gaps by strain is well established, for instance in quantum wells. This provides a mechanism for chain-length disorder to induce gap fluctuations. We have added a rough quantitative estimate of this effect based on hybrid DFT calculations of the CdSe band gap under tensile strain (Supplementary Figure S11), which show that the strain from oleic acid ligands induces a gap shift of 0.14 eV in 4ML platelets, comparable to but somewhat smaller than our calculated ligand-type shift of 0.237 eV. However, we believe this is still an underestimate of the effect of chain-length fluctuations. Firstly, the "oleate-passivated" platelets in Ref 34 likely still contain some acetate ligands, so the strain would be larger in a pure-oleate nanoplatelet. Secondly, while the lattice strain remains very small and thus nearly isotropic in acetate-passivated platelets, long carboxylate ligands will lead to strongly anisotropic strain rotated by 90 degrees on the top and bottom surfaces. This anisotropy is averaged out in the experimental diffraction measurements, so the distortion due to the ligands and the resulting α are underestimated. Therefore, the effect of ligand chain-length disorder on the linewidth can be expected to be similar to that of ligand-type disorder. Since a full treatment of chain-length disorder is beyond the scope of the present manuscript, we have elected to keep the focus of the rest of the manuscript on ligand-type disorder.

With this additional information, we believe that we have justified ligand type disorder as a reasonable proxy for all types of ligand disorder, including chain-length disorder, and therefore our claim that this work may solve the "linewidth puzzle" is well supported.

Reviewer #3 (Remarks to the Author):

The authors have modified their manuscript, taking into account my comments.

I could suggest citing other papers on the ligand-induced deformation of nanoplatelets apart from those of Ithurria's group but i will let the authors decide whether this is appropriate:

[10.1021/acs.chemmater.7b05324](https://doi.org/10.1021/acs.chemmater.7b05324)

[10.1073/pnas.2316299121](https://doi.org/10.1073/pnas.2316299121)

[10.1126/sciadv.1701483](https://doi.org/10.1126/sciadv.1701483)

I am now happy to recommend the paper for publication. I would like to thank the authors for their engagement during the peer review process, which I think helped the paper reach higher standards.